

# Computational inference of H3K4me3 and H3K27ac domain length

Julian Zubek[1,2], Michael L. Stitzel[3,4], Duygu Ucar[3,4] and Dariusz M. Plewczynski[1,5]

[1] Centre of New Technologies, University of Warsaw, Warsaw, Mazovia, Poland
[2] Institute of Computer Science, Warsaw, Mazovia, Poland
[3] Institute for Systems Genomics, Univeristy of Connecticut, Farmington, CT, United States of America
[4] The Jackson Laboratory for Genomic Medicine, Farmington, CT, United States of America
[5] Faculty of Pharmacy, Medical University of Warsaw, Warsaw, Poland

## ABSTRACT

**Background.** Recent epigenomic studies have shown that the length of a DNA region covered by an epigenetic mark is not just a byproduct of the assaying technologies and has functional implications for that locus. For example, expanded regions of DNA sequences that are marked by enhancer-specific histone modifications, such as acetylation of histone H3 lysine 27 (H3K27ac) domains coincide with cell-specific enhancers, known as super or stretch enhancers. Similarly, promoters of genes critical for cell-specific functions are marked by expanded H3K4me3 domains in the cognate cell type, and these can span DNA regions from 4–5kb up to 40–50kb in length. These expanded H3K4me3 domains are known as buffer domains or super promoters.
**Methods.** To ask what correlates with—and potentially regulates—the length of loci marked with these two important histone marks, H3K4me3 and H3K27ac, we built Random Forest regression models. With these models, we computationally identified genomic and epigenomic patterns that are predictive for the length of these marks in seven ENCODE cell lines.
**Results.** We found that certain epigenetic marks and transcription factors explain the variability of the length of H3K4me3 and H3K27ac marks across different cell types, which implies that the lengths of these two epigenetic marks are tightly regulated in a given cell type. Our source code for the regression models and data can be found at our GitHub page: https://github.com/zubekj/broad_peaks.
**Discussion.** Our Random Forest based regression models enabled us to estimate the individual contribution of different epigenetic marks and protein binding patterns to the length of H3K4me3 and H3K27ac deposition patterns, therefore potentially revealing genomic signatures at cell specific regulatory elements.

## INTRODUCTION

Epigenomics refer to the heritable changes that are not stemming from the changes in the genomic DNA sequence. The major epigenetic mechanisms can be categorized into two distinct classes: (1) methylation of the cytosine residues of DNA (i.e., DNA methylation); and (2) the post-translational modifications of the histone proteins

Corresponding authors
Duygu Ucar, Duygu.Ucar@jax.org
Dariusz M. Plewczynski,
dariuszplewczynski@gmail.com,
d.plewczynski@cent.uw.edu.pl

(i.e., histone modifications). Recent advancements in next generation sequencing (NGS) technologies have enabled the development of experimental methods to generate genome-wide profiles of epigenetic marks in many types of cells and organisms. These technologies enable profiling epigenetic states genomewide as collections of short sequence reads associated with a given epigenetic mark, which is then aligned to the reference genome. Each locus in the genome has therefore an associated read count, where regions with high abundance of reads (i.e., peaks) are captured to identify loci marked by the profiled epigenetic mark. It has been previously shown that the length of a particular locus covered by an epigenetic mark is related to the cell-specific functions of that region. For example, several research groups observed that long genomic segments marked by enhancer-specific histone modifications, such as acetylation of histone H3 lysine 27 (H3K27ac), are found at cell-specific enhancer loci (*Hnisz et al., 2013*; *Chapuy et al., 2013*; *Parker et al., 2013*). Similar observations have been made for DNA methylation profiles. Continuous regions of low DNA methylation (at least five kilobase (kb) in length), namely DNA methylation valleys or canyons, have been associated with transcription factors and with genes known to regulate development in embryonic stem cells and genes with potential involvement in the regulation of hematopoiesis in hematopoietic stem cells (HSCs) (*Jeong et al., 2014*). In addition, in a recent publication, we showed that longer domains (from 5 to 50 kb) of histone H3 lysine 4 trimethylation (H3K4me3) preferentially mark genes associated with cell identity and function in diverse cell types and organisms (*Benayoun et al., 2014*). These expanded H3K4me3 domains, termed broad domains, have also been shown to mark tumor suppressors in normal cells when compared to tumor pairs (*Chen et al., 2015*).

   These recent studies show that the long continuous segments of DNA covered by H3K4me3 and H3K27ac can be used as an epigenetic signature for identifying cell-type-specific promoters and enhancers and these epigenetic patterns might be established and maintained by the cell to robustly regulate cell-type specific gene expression patterns and functions. However, what potentially regulates the deposition or maintenance of such cell-specific epigenetic signatures still remains an open question. Computationally identifying genomic and epigenomic characteristics associated with the length of these histone marks might provide some hints about potential mechanisms. For this purpose, to systematically identify genomic and epigenomic characteristics associated with cell-specific epigenetic signatures, we build Random Forest regression models to explain the length of H3K4me3 and H3K27ac domains using both genomic and epigenomic information. Our regression models are different than our previous method that is mainly comparing very long H3K4me3 domains to shorter domains (*Benayoun et al., 2014*). Here, we study the length of a domain as a spectrum without arbitrarily introducing a length cut-off. Moreover, we also study H3K27ac patterns in addition to H3K4me3, which is the mark that has been used for identifying cell-type-specific enhancers. To the best of our knowledge, this is the first attempt to study the length of an epigenetic mark under the light of other genomic and epigenomic characteristics of that site. A recent algorithm showed that in the existence of many epigenetic marks, missing epigenetic marks could be imputed at 25-bp resolution using regression tress (*Ernst & Kellis, 2015*). Although this method and study was informative in showing that epigenetic marks are not independently deposited from

each other, it did not reveal whether any of these correlations are important for the length of H3K4me3 and H3K27ac domains. Moreover, this method was restricted to histone modification profiles and didn't study the relationship between protein binding patterns and the length of these two important epigenetic marks.

Using our regression models in seven ENCODE cell lines (human embryonic stem cell lines H1 and H9 and cancer cell lines K562, HCT116, GM12878, A549 and HeLa), we asked whether the length of loci marked with H3K4me3 or H3K27ac domains can be predicted using other genomic features of that loci including transcription factor (TFs) binding patterns and other epigenetic marks using chromatin immunoprecipitation followed by sequencing (ChIP-seq) datasets. This enabled us to systematically study the relationship between H3K4me3 and H3K27ac length found at a locus and other genomic features of that locus. Moreover, we also quantified the importance of different genomic features in describing the epigenetic domain length. Our analyses revealed that with integrative computational models we could predict the length of H3K4me3 and H3K27ac domains with high efficacy (>0.9 correlation on the average between observed and predicted). Moreover, with our models we can infer the most important genomic and epigenomic features that are predictive for the length of these two epigenetic marks.

## MATERIALS & METHODS

**ChIP-seq data pre-processing:** Publicly available datasets were obtained from ENCODE consortium (*Hoffman et al., 2013*), in multiple ENCODE cell lines as well as in pancreatic islets (accession numbers are listed in Table S1). To preprocess ChIP-seq datasets, we mapped reads to reference human genome (hg19) using bowtie0.12.7 (*Langmead et al., 2009*). ChIP-seq peaks were called using MACS2.08 (*Jeong et al., 2014*) with the "—broad" option for histone marks. Peaks were assigned to the gene with the closest transcription start site (TSS).

### Domain length prediction using Random Forest regression algorithm

**Feature extraction:** For each histone modification domain and data feature being studied, we used the fraction of overlapping basepairs (bps) in our models to eliminate the bias that could be introduced by differences in domain lengths. For example, if an H3K4me3 domain has length 100 base pairs and 15 of these bases also have CHD1 binding, fraction of overlap for that domain for the CHD1 feature is $(15/100 = 0.15)$. Note that a single H3k4me3 domain can simultaneously overlap with multiple instances of the same feature (e.g., multiple CHD1 binding sites within a domain). For these instances, we used the sum of all of such overlaps. For example, as depicted in the Fig. 1C domain B overlaps with two instances of domains A, therefore the final score for this domain for feature A is $=$ sum(bp_A)/length(B).

**Model building:** Random Forest regression with 100 trees was performed on the constructed data sets. We employed these models to unfold the relationship between domain length and the constructed features. We used an implementation available in scikit-learn library (*Pedregosa et al., 2011*). Gini impurity was used as a quality measure

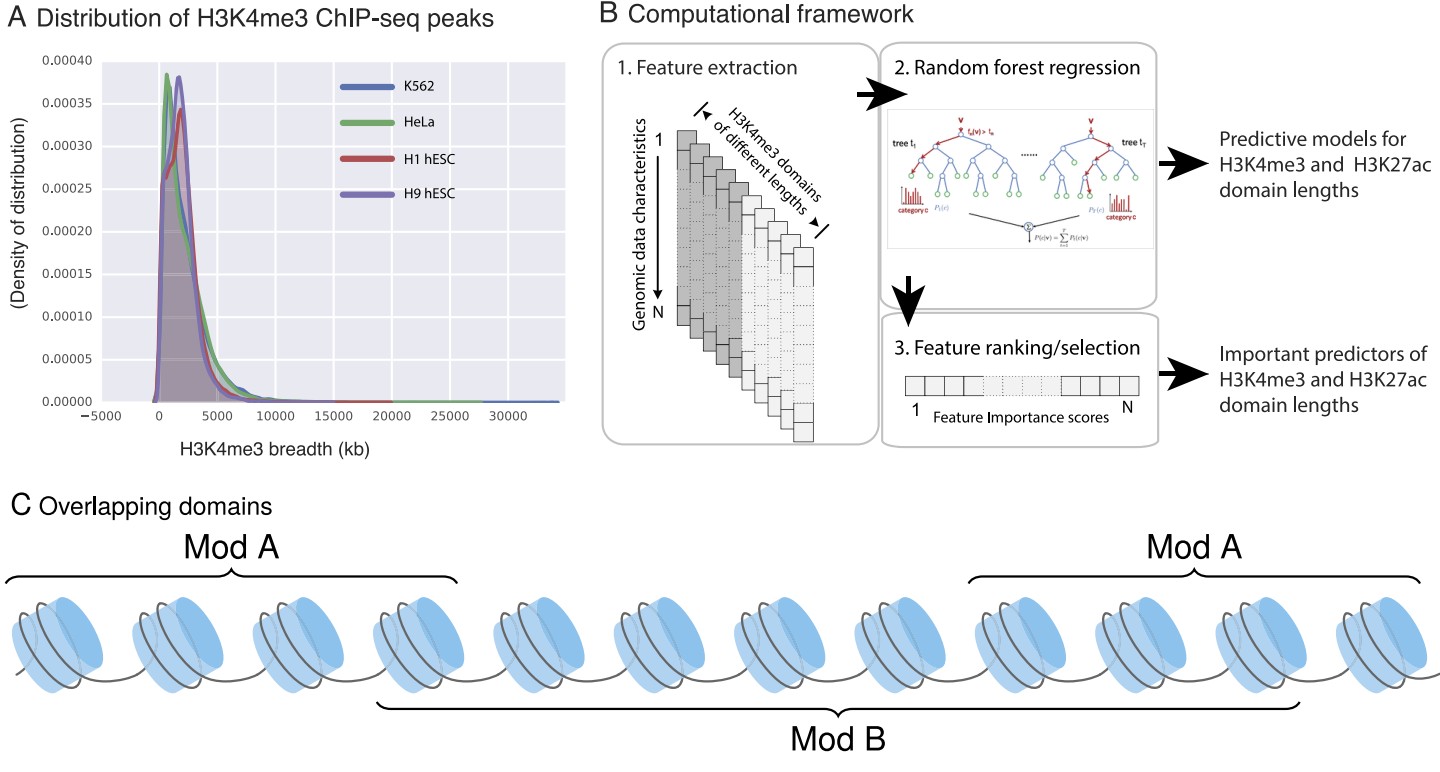

**Figure 1  H3K4me3 domain lengths vary.** (A) H3K4me3 domain length distribution in four ENCODE cell lines. H3K4me3 domain lengths span a wide range and longer domains mark cell-type-specific promoters. (B) Our three-step framework to model the length of H3K4me3 and H3K27ac domains. (C) An example of domain type B, which overlaps with two instances of domains of type A.

for choosing best splits. At each level $\sqrt{m}$ randomly chosen attributes were considered as candidates for split (where $m$ is the number of all attributes).

**Feature ranking:** Feature rankings enable identifying and ranking important features for the regression problem at hand. When building a decision tree, the feature that leads to the greatest decrease in Gini impurity score is chosen at each split. Importance score for each feature is the mean decrease of impurity for all tree nodes, which is averaged over all trees in the ensemble. Calculating importance scores enable comparison of attributes, but does not state the significance of importance scores. To assess significance we employed a Monte-Carlo technique based on contrast attributes, which are random permutations of the original attributes. To quantify the significance of a feature importance score, we followed the below procedure:

1. For each of the original attributes its values were permuted at random and added as a new contrast attribute to the original dataset.
2. Random Forest model was trained on the dataset consisting of both original and contrast attributes.
3. Standard deviation (SD) of importance scores for contrast attributes was calculated. The value equal to $2 \times$ SD was used as a cutoff for minimal significant difference between importance scores.

4. Feature importance scores were sorted in decreasing order. Differences between subsequent scores were calculated. We looked for the last pair of subsequent scores for which the difference was larger than the cutoff value. The larger of the scores from that pair constituted a threshold for dividing features into significant and not significant.

This procedure enabled us to identify a small set of significant features for our models.

## RESULTS

### H3K4me3- and H3K27ac-domain lengths are informative for cell-type specificity

It has been observed that epigenetic marks can decorate loci with domains of varying length. For example, when we study the domain lengths for H3K4me3, we observe that the length distribution is not uniform and the length of these peaks vary between a few hundred and 20 kb as shown in the length distribution of H3K4me3 domains in H1 human embryonic stem cells (hESCs) (Fig. 1A). The span of epigenetic mark deposition has gained attention in recent years with several studies, including ours, showing that particularly long stretches of DNA marked with H3K4me3 or H3K27ac specifically coincide with cell-type specific promoters or enhancers, respectively (*Hnisz et al., 2013*; *Chapuy et al., 2013*; *Parker et al., 2013*; *Benayoun et al., 2014*; *Bernstein, Meissner & Lander, 2007*).

Given the association of large domains with functionally important DNA elements, we built a computational framework that can (i) assess whether—and to what extent—the length of an H3K4me3 (or H3K27ac) domain at a locus can be predicted from other genomic and epigenomic characteristics of that locus and (ii) quantify the ability of each genomic and epigenomic characteristic to predict the length of domains of these two marks. Our framework consisted of three stages (summarized in Fig. 1B): (1) extracting genomic and epigenomic characteristics of each domain; (2) building a regression model (based on Random Forest regression) for the length of the domains as a function of other data characteristics; and (3) delineating the predictive genomic signature for domain length by prioritizing and selecting important predictors. The signatures we obtained will help us to identify candidate molecular mechanisms for setting or maintaining domain length of this epigenetic mark.

### H3K4me3 and H3K27ac deposition lengths can be predicted with high accuracy by integrating other genomic datasets

For each H3K4me3 or H3K27ac domain, we extracted other genomic and epigenomic characteristics from ENCODE datasets. We used the fraction of overlap (as explained in methods) to normalize feature overlaps based on domain length, so longer domains would not have an inflation of overlapping features. Next, in seven ENCODE cell lines (H1 and H9 hESCs, K562, HCT116, GM12878, A549, HeLa) as well as in human pancreatic islets, we built Random Forest Regression models to predict the length of H3K4me3 and H3K27ac domains. Our analyses showed that these models can predict the length of epigenetic marks with high accuracy (>0.9 correlation between observed and predicted lengths; Fig. 2).
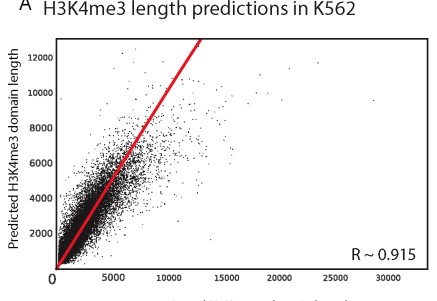

A  H3K4me3 length predictions in K562

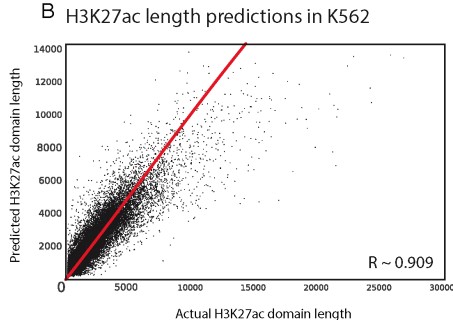

B  H3K27ac length predictions in K562

C  Correlation table for all cell types

| Cell type | H3K4me3 breadth prediction | | H3K27ac breadth prediction | |
|---|---|---|---|---|
| | Pearson R | Spearman R | Pearson R | Spearman R |
| K562 | 0.885985 | 0.914721 | 0.893642 | 0.908684 |
| HCT116 | 0.862842 | 0.891768 | 0.873392 | 0.900026 |
| GM12878 | 0.911673 | 0.909762 | 0.878918 | 0.895061 |
| A549 | 0.887127 | 0.920998 | 0.852752 | 0.897962 |
| HeLa | 0.882812 | 0.911711 | 0.863694 | 0.864861 |
| H1 | 0.871091 | 0.899682 | 0.859651 | 0.847423 |
| H9 | 0.783717 | 0.797553 | 0.658957 | 0.703986 |

**Figure 2** **Our computational models can predict the length of H3K4me3 and H3K27ac domains with high precision.** (A) Predicted vs. observed H3K4me3 domains using our model in K562. (B) Predicted vs. observed H3K27ac domains using our model in K562. (C) Correlation between predicted and observed domain lengths in all studied cell types.

## Conserved genomic features are predictive for H3K4me3 and H3K27ac deposition lengths

Random Forest quantifies the importance of each feature by calculating its contribution to the overall classification accuracy (see methods for details). We calculated feature importance scores for each genomic and epigenomic data type we used in our models. Next, we assessed the significance of these scores by comparing them with scores produced by random permutations of the original features (i.e., contrast features). With this methodology, we identified the significant predictors of length for H3K4me3 and H3K27ac in each studied cell type. For example, we showed that in K562 CHD1 is one of the most important predictors for both H3K4me3 and H3K27ac domains (Figs. 3 and 4).

Our random forest regression models revealed that there are frequently repeating features that are predictive for both H3K4me3 and H3K27ac epigenetic marks, which point out to putative upstream or downstream associations between the length of epigenetic marks and these predictive features. Certain genomic and epigenomic datasets frequently rank as important for predicting the length of these two marks across cell types (Figs. 3C–3D). For H3K4me3, CHD1- and Pol2-binding and the H3K79me2 histone modification are the most important predictors as assessed by feature importance scores. For H3K27ac, TEAD1- and CHD1-binding and the presence of other acetylation marks (H3K4ac and H2AK5ac) are important genomic characteristics.

## Elongated CHD1 binding might promote robust and increased expression patterns

Among the top predictors, CHD1 is an H3K4me3-associated chromatin remodeler and it is shown to be essential for the expression of key developmental genes in mouse embryonic stem (ES) cells as well as for reprogramming of fibroblasts to the pluripotent stem cells (*Gaspar-Maia et al., 2009*). Moreover, CHD1 binding is shown to be essential for the assembly of preinitiation complexes (PIC), which is necessary to start transcription. PIC includes co-activators like Mediator complex and the TFIID complex, along with Pol2 and the general transcription factors (*Lin et al., 2011*). Given the importance of CHD1 in regulating PIC assembly and ultimately gene expression, we have conducted further analyses to study the relationship between CHD1 binding and the length of H3K27ac

## A  H3K4me3 length predictors in K562

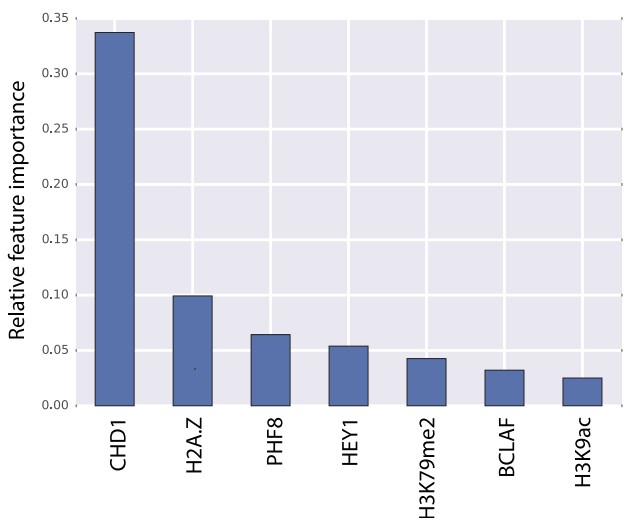

## B  H3K27ac length predictors in K562

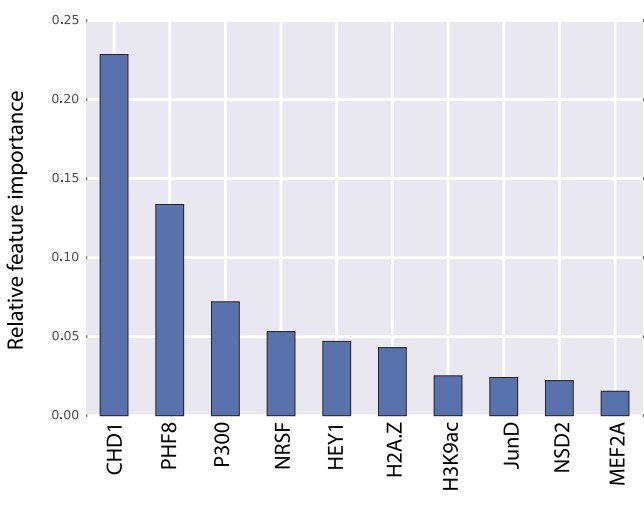

## C  Feature importance scores (H3K4me3)

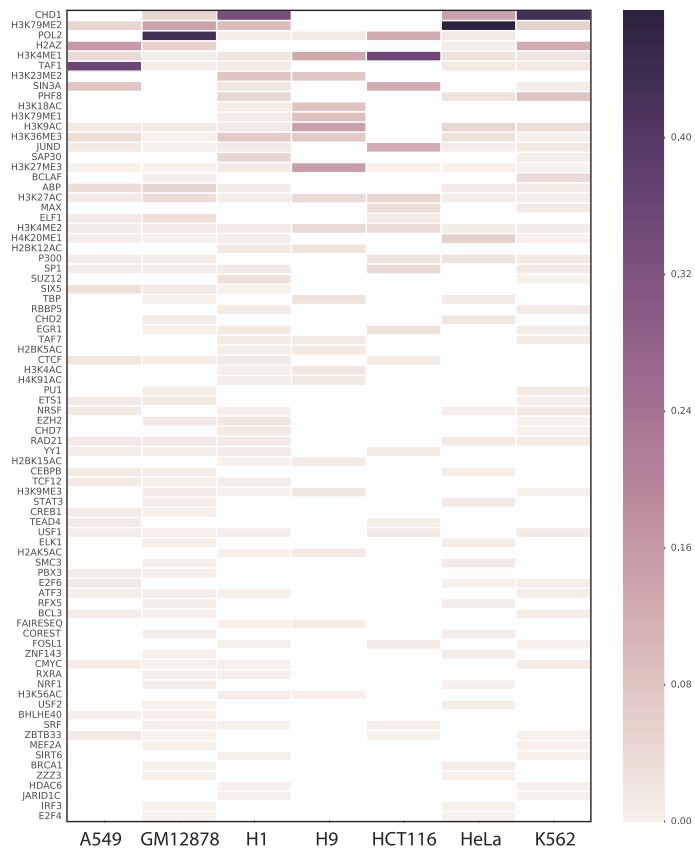

## D  Feature importance scores (H3K27ac)

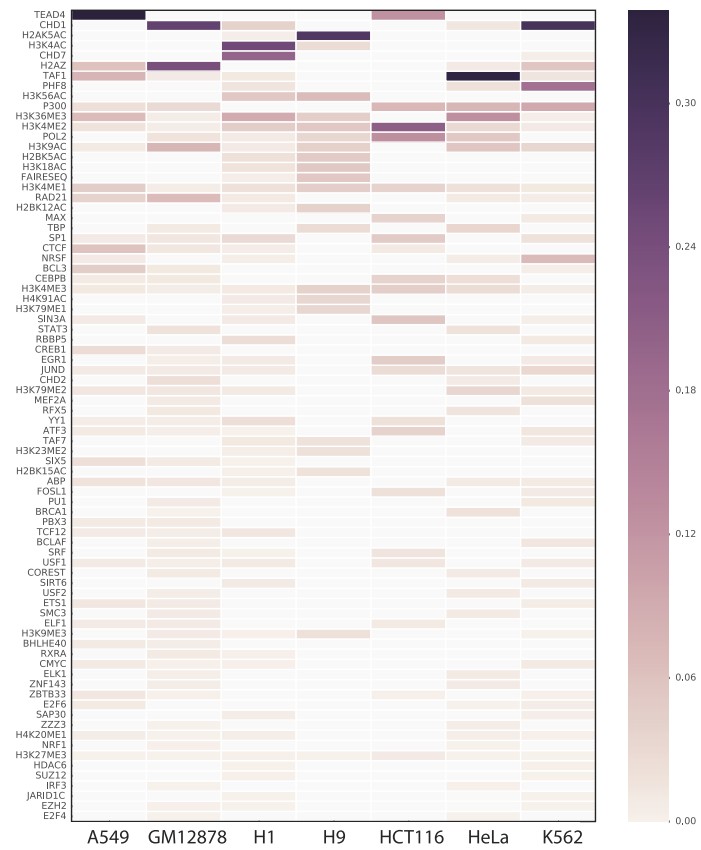

**Figure 3  We identified genomic features that play an important role in H3K4me3 and H3K27ac domain lengths.** Significant predictors of H3K4me3 domain length in K562 (A) and in all studied cell types (C). Significant predictors of H3K27ac domain length in K562 (B) and in all studied cell types (D). Missing datasets in the studied cell types are represented in white.
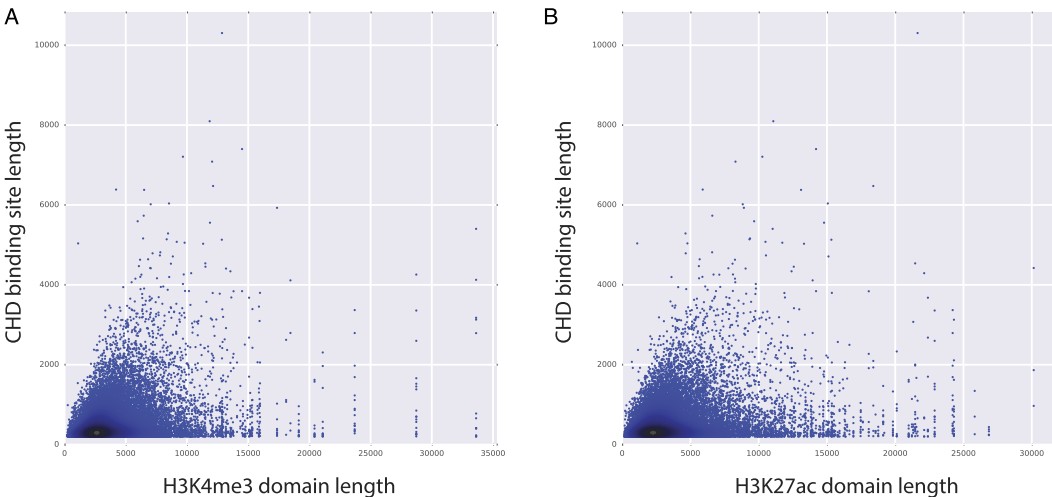

**Figure 4** **CHD1 binding length and H3K4me3 (and H3K27ac) domain lengths are correlated.** *X* axes represents the length of H3K4me3 domains (A) and the length of H3K27ac domains (B). *Y* axes represents the length of CHD1 bound loci.

and H3K4me3 domains. Our analyses showed that there is a positive correlation between the length of CHD1 and the lengths of both H3K27ac and H3K4me3 domains (Fig. 4), implying that elongated CHD1 binding takes place at regions where longer H3K4me3 and longer H3K27ac deposition is observed. Broad H3K4me3 domains have been associated with increased and robust expression of genes, similarly genes marked by longer H3K27ac domains (i.e., super/stretch enhancers) have been associated with increased expression. Our findings show that longer H3K4me3 and H3K27ac domains have longer deposition of CHD1 binding; which might imply a more robust assembly of mediator complex and PIC enabled by the elongated binding of CHD1 protein. This robust assembly of PIC at these loci might explain their robust and increased expression patterns.

## Broader H3K4me3 and H3K27ac are bounded by CTCF mediated interactions

CTCF, is a TF that is important for the insulator activity. Insulators are boundary elements that restrict the regulatory interactions between enhancers and promoters within a boundary element. An intriguing hypothesis based on the high predictive value of CTCF for the length of H3K4me3 and H3K27ac domains is that the longer H3K27ac and H3K4me3 domains tend to be enriched at the boundary of chromatin domains marked by the CTCF binding. To test this, we analyzed CTCF mediated chromatin interactions obtained by the Chromatin Interaction Analysis by Paired-End Tag Sequencing (ChIA-PET) technology in GM12878 cell line (*Tang et al., 2015*). We split H3K4me3 and H3K27ac domains into five bins based on their domain length and for each of these bins, we studied the localization of CTCF mediated chromatin interactions with respect to the center of the domain overlapping the CTCF fragment. Our analyses revealed an interesting pattern, where we observed that with increasing length of H3K27ac and H3K4me3 domains, CTCF interactions tend to cluster at the beginning or at the end of the domains (Fig. 5), which

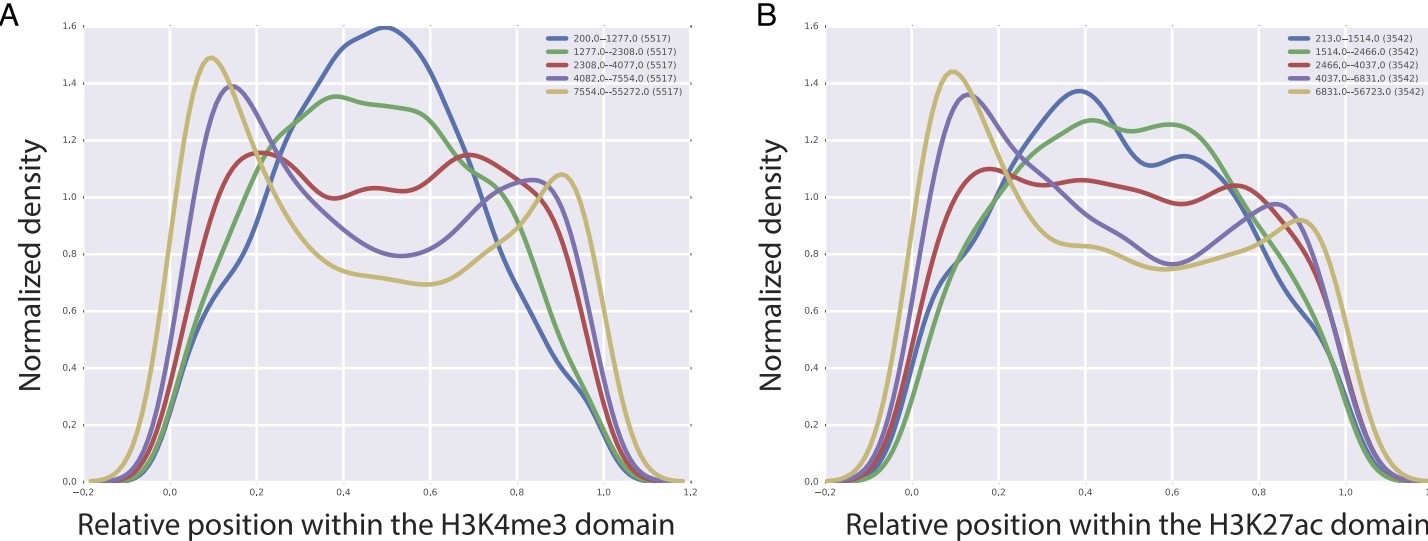

**Figure 5** **Localization of CTCF interactions over H3K4me3 (A) and H3K27ac (B) domains.** *X* axis represents the relative position on the domains. *Y* axis represents the CTCF interaction coverage. Yellow represents the longest domains, blue represents the shortest domains.

implies that longer H3K4me3 and H3K27ac domains tend to be at the boundary of CTCF defined boundary domains.

## DISCUSSION

Recent epigenomic studies, including ours, have shown that the length of certain epigenetic mark deposition domains—H3K4me3 for promoters and H3K27ac for enhancers—might have functional implications and can be predictive in identifying promoters and enhancers that are critical for the underlying cell's identity and function. However, we still do not know the mechanisms that establish and maintain the boundaries of these domains, therefore it is still intriguing to investigate whether these domain lengths correlated with other genomic and epigenomic datasets using regression models, which might be important players in setting and maintaining these domains. Our analyses showed that in fact these domain lengths could be predicted with high accuracy using publically available Transcription Factor (TF) binding and histone modification ChIP-seq datasets in seven ENCODE cell lines as well as in human pancreatic islet samples. This high accuracy of our regression models show that the length of these domains correlates well with the genomic and epigenomic states of these sites within a cell and not a byproduct of assays or analyses. We conducted further analyses to understand in which ways these important predictors might be related to broad H3K4me3 and H3K27ac domains. Our analyses revealed that the nature of the relationship is based on the genomic feature in consideration. Our analyses with CHD1 binding revealed that elongated CHD1 binding is associated with longer H3K27ac and H3K4me3 domains, which might contribute to the robust and increased expression of genes associated with these domains via a robust assembly of PIC. On the other hand, for CTCF domains, we noticed that broader H3K4me3 and H3K27ac domains tend to locate towards the edges of CTCF defined boundary elements. Although our

computational analyses cannot infer causality in the establishment of CTCF domains and broader deposition of epigenetic marks, we still observe that the deposition of domain length is not independent of the regulatory boundaries defined by CTCF binding.

## CONCLUSIONS

Our analyses revealed that there are frequently occurring predictors for H3K4me3 and H3K27ac domain lengths. Among the predictors we identified, CHD1-binding is important for the prediction of both H3K27ac and H3K4me3 domain lengths. CHD1 is essential for the assembly of preinitiation complexes (PIC) and ultimately for regulating gene expression levels. We have conducted further analyses to show that broader H3K4me3 and H3K27ac domains have elongated CHD1 binding, which might play a role in their robust and increased expression patterns. In the future, it would be interesting to study the effect of CHD1 knockdown or precise targeting on the expression levels of genes marked by broad H3K4me3 and H3K27ac domains. We also showed that broader H3K4me3 and H3K27ac domains localize towards the end of the regulatory boundary elements defined by CTCF binding by studying chromatin interactions datasets generated by ChIA-PET technology. In summary, our computational analyses revealed interesting features of epigenetic mark length for two important histone modification marks (H3K4me3 and H3K27ac) and showed in which ways these signals correlate with TF binding patterns and chromatin interactions.

### Funding

This article is funded by the European Union from financial resources of the European Social Fund, Project PO KL "Information technologies: Research and their interdisciplinary applications"; 2015/16/T/ST6/00493, 2014/15/B/ST6/05082 and 2013/09/B/NZ2/00121 grants from the Polish National Science Centre; COST BM1405 and BM1408 EU actions. The funders had no role in study design, data collection and analysis, decision to publish, or preparation of the manuscript.

### Grant Disclosures

The following grant information was disclosed by the authors:
European Union.
Polish National Science Centre: 2015/16/T/ST6/00493, 2014/15/B/ST6/05082, 2013/09/B/NZ2/00121.
EU actions: COST BM1405, BM1408.

### Competing Interests

The authors declare there are no competing interests.

## Author Contributions

- Julian Zubek performed the experiments, analyzed the data, contributed reagents/materials/analysis tools, wrote the paper, prepared figures and/or tables, reviewed drafts of the paper.
- Michael L. Stitzel wrote the paper, reviewed drafts of the paper.
- Duygu Ucar conceived and designed the experiments, performed the experiments, analyzed the data, contributed reagents/materials/analysis tools, wrote the paper, prepared figures and/or tables, reviewed drafts of the paper.
- Dariusz M. Plewczynski conceived and designed the experiments, analyzed the data, wrote the paper, reviewed drafts of the paper.

## Data Availability

http://zubekj.github.io/broad_peaks/.

## Supplemental Information

Supplemental information for this article can be found online at http://dx.doi.org/10.7717/peerj.1750#supplemental-information.

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
