# Peer review of "Computational inference of H3K4me3 and H3K27ac domain length"

_PeerJ, doi:10.7717/peerj.1750_

## Round 0.1 · original submission · Major Revisions

· Academic Editor

Major Revisions

Both reviewers request important clarifications in the introduction, specifically regarding (Reviewer #1) the diffferences between this manuscript and the earlier Benayoun et al. work, and on the relevance of the computational approach (Reviewer #2) in the light of the recent publication of ChromImpute. Both reviewers require more thorough discussion of the biological significance of the results and comparison with results by other researchers.

My own comments on the manuscript:

Although the construction of the model is described, the model itself is not provided and this prevents its use by other workers. The model (either as a script, a stand-alone program, a GitHub link, website, etc) should be made available, as required by PeerJ policy. https://peerj.com/about/policies-and-procedures/#data-materials-sharing

The first panel in Figure 1 seems to be identical with panel A from Fig 1 in 10.1016/j.cell.2014.06.027 , which has been co-authored by one of the present authors. Although this panel is merely illustrative and not intended to represent new data, it would be better to replace it to avoid any possibility of misunderstanding.

Since PeerJ is a journal with a broad readership, I would advise authors to define common molecular biology abbreviations (like TSS for transcription start site) at their first mention in the text.

For your convenience, I have tracked down the full references to the papers mentioned by Reviewer #2, which seem to conflict with some of your results. These are:
Lee et al. (2012) 10.1101/gad.186841.112
Thornton 2013 (actually 2014) 10.1101/gad.232215.113

Reviewer 1 ·

Basic reporting

No Comments

Experimental design

PeerJ requires that the submission must describe original primary research.
The work presented in this manuscript seems very close to part of the work in a recent publication ( Benayooun, et al. 2014, Cell). They used the same method (Random Forest model) to investigate the same topic (identify chromatin features that contribute to the prediction of H3K4me3 domain length) and reached to very similar results. See their figure 4 and related supplemental figures. It may be important to clarify the difference in as much detail as possible and add this clarification to the manuscript.

Validity of the findings

PeerJ requires that the data should be robust.
For the chromatin features that appear to be the best predictor, in what manner are they associated with domain length of H3K4me3 or H3K27ac? A recent publication (Chen, et al. 2015, Nature Genetics) reported that most of the active marks in promoter or gene body appear to be broad at genes associated with broad H3K4me3 (see their Figure 2 and supplemental figures). Could the authors also do similar analysis for the chromatin features they investigated? For example, does H3K4me3 or H3K27ac domain length correlates with peak width (or height) of CHD1 and other good predictor? This type of data will make the observation much more meaningful, convincing, and thus robust.

PeerJ requires that the data should be controlled.
For the chromatin features that appear to be the best predictors, are they also good predictors for peak height or peak total signal of H3K4me3 or H3K27ac?

Additional comments

The manuscript presented a cutting-edge research. Scientists discovered very recently that the domain length of active mark, such as H3K4me3 or H3K27ac, on chromatin has important biological implication. It is still a mystery how the domain length is determined, and why the broad domain is associated with certain category of gene functions, such as cell identity maintenance or tumor suppression. The authors of this manuscript used Random Forest algorithm to define chromatin features that are associated with domain length of H3K4me3 or H3K27ac. The research has revealed some very interesting observations. The manuscript may become even better if it can present some biological mechanisms underlying the observations.

Reviewer 2 ·

Basic reporting

There are already a significant number of publication regarding feature selection using histone modification, transcription factors and their co-factors. In this manuscript, the authors suggested to predict the length of H3K4me3 and H3K27ac compared to other previous feature selection models.

However, the review doesn’t see clear reason to predict the length of H3K4me3 and H3K27ac. The purpose has already been implemented more extensively by ChromImpute.

Potential causal effect of CHD1 and H3K4me3 could be interesting. There are a couple of H3K4me3 ChIPseq dada after Chd1 knockdown. Still, however, previous study found that Chd1 has no effect to H3K4me3 (Lee et al, 2012, G & D; Thornton et al, 2013, G&D). The previous articles might not check the length of H3K4me3. The data should be further processed by the authors to support their findings. If Chd1 does not cause H3K4me3, what could be the interpretation?

CTCF, an insulator, is one of the important features. What could be biological interpretation for this?

For H3K27ac, TEAD was found as a major factor. It must be cell-type specific, considering that it is not on the top list in K562. What could be cell-type specific or constitutive marks for the cell types that authors studied?

Experimental design

No comments

Validity of the findings

No comments

Additional comments

No comments

---

## Round 0.2 · accepted · Accept

· Academic Editor

Accept

The manuscript has been improved to a high standard. I am glad to approve it.

Reviewer 1 ·

Basic reporting

No Comments

Experimental design

No Comments

Validity of the findings

No Comments

Additional comments

All my concerns has been well addressed. Thanks!

Reviewer 2 ·

Basic reporting

The manuscript has been significantly improved by answering the reviewer's comment. Especially, the observation of CTCF at the boundary of broad H3K4me3 and H3K27ac domain is interesting.

Experimental design

N/A

Validity of the findings

N/A